::ᐧᐧᐧ PLOS | ONE

# Use of diuretics in shock: Temporal trends and clinical impacts in a propensity-matched cohort study

Ghassan Bandak[1], Ankit Sakhuja[2], Nicole M. Andrijasevic[3], Tina M. Gunderson[4], Ognjen Gajic[5], Kianoush Kashani [5,6]*

1 Division of Pulmonary and Critical Care Medicine, Marshall Health, Huntington, WV, United States of America, 2 Division of Pulmonary and Critical Care Medicine, University of West Virginia, Morgantown, WV, United States of America, 3 Department of Anesthesiology and Perioperative Medicine, Mayo Clinic, Rochester, Minnesota, United States of America, 4 Division of Biomedical Statistics and Informatics, Mayo Clinic, Rochester, Minnesota, United States of America, 5 Division of Pulmonary and Critical Care Medicine, Mayo Clinic, Rochester, Minnesota, United States of America, 6 Division of Nephrology and Hypertension, Mayo Clinic, Rochester, Minnesota, United States of America

* kashani.kianoush@mayo.edu

**Data Availability Statement:** All relevant data are in the manuscript and its Supporting Information files.

## Abstract

### Objective

Fluid overload is common among critically ill patients and is associated with worse outcomes. We aimed to assess the effect of diuretics on urine output, vasopressor dose, acute kidney injury (AKI) incidence, and need for renal replacement therapies (RRT) among patients who receive vasopressors.

### Patients and methods

This is a single-center retrospective study of all adult patients admitted to the intensive care unit between January 2006 and December 2016 and received >6 hours of vasopressor therapy and at least one concomitant dose of diuretic. We excluded patients from cardiac care units. Hourly urine output and vasopressor dose for 6 hours before and after the first dose of diuretic therapy was compared. Rates of AKI development and RRT initiation were assessed with a propensity-matched cohort of patients who received vasopressors but did not receive diuretics.

### Results

There was an increasing trend of prescribing diuretics in patients receiving vasopressors over the course of the study. We included 939 patients with median (IQR) age of 68(57, 78) years old and 400 (43%) female. The average hourly urine output during the first six hours following time zero in comparison with average hourly urine output during the six hours prior to time zero was significantly higher in diuretic group in comparison with patients who did not receive diuretics [81 (95% CI 73–89) ml/h vs. 42 (95% CI 39–45) ml/h, respectively; p<0.001]. After propensity matching, the rate of AKI within 7 days of exposure and the need for RRT were similar between the study and matched control patients (66 (15.6%) vs. 83 (19.6%), p = 0.11, and 34 (8.0%) vs. 37 (8.7%), p = 0.69, respectively). Mortality, however,

**Funding:** This work was supported by a small Grant for statistical support from Critical Care research committee at Mayo Clinic. The funder had no role in study design, data collection and analysis, decision to publish, or preparation of the manuscript.

**Competing interests:** The authors have declared that no competing interests exist.

was higher in the group that received diuretics. Ninety-day mortality was 191 (45.2%) in the exposed group VS 156 (36.9%) p = .009.

## Conclusions

While the use of diuretic therapy in critically ill patients receiving vasopressor infusions augmented urine output, it was not associated with higher vasopressor requirements, AKI incidence, and need for renal replacement therapy.

## Introduction

Fluid management is one of the cornerstones of critically ill patient's management [1, 2]. Early fluid resuscitation has been shown to improve outcomes in critically ill patients [2]. The spectrum of volume management in critically ill patients has been described in four distinct stages starting with resuscitation or salvage phase focusing on maintaining perfusion and cardiac output, followed by the optimization phase which uses targeted fluid therapy to improve oxygen delivery, then a stabilization phase with a focus on preventing further organ damage, and lastly the de-escalation phase where patients are weaned off support and achieve negative fluid balance [3–5].

Over two-thirds of critically ill patients qualify the definition of volume overload (i.e., increase the weight more than 10% of admission body weight) in their first day of intensive care unit (ICU) stay [6], and most patients will be discharged from ICU while volume overloaded [7]. Fluid overload is associated with increased mortality in different patient populations, such as patients with sepsis, acute lung injury (ALI), and acute kidney injury (AKI) [8–14]. Not only volume overload, but the length of the remaining volume overloaded is associated with worse outcomes [13]. Furthermore, fluid overload was correlated with an increased need for medical interventions [6], including the need for renal replacement therapy, mechanical ventilation [9], and decreased mobility [7].

The use of diuretic therapy among critically ill patients was investigated in different studies with mixed results. Mehta and colleagues showed increased mortality and non-recovery of renal function associated with the use of diuretics [15]. Other studies reported no association between diuretic use and improved survival [16–18], or renal recovery rates. Grams and colleagues suggested that the use of diuretics may improve mortality rates in patients with ALI and AKI [11]. In the trauma population, the use of furosemide in fluid overloaded patients was associated with better volume control and no harmful effects on hemodynamics [19].

We believe there is equipoise regarding the initiation of the fluid de-escalation phase of resuscitation while patients are still on vasopressors. Therefore, we designed a historical cohort study with propensity matching to evaluate the effect of diuretic utilization among patients who are in a steady dose of vasopressors. We hypothesized that the use of diuretics in patients receiving vasopressor infusion will increase urine output, will not negatively affect the hemodynamic status, and will not lead to worsening kidney function. We also assessed the incidence of AKI and the need for renal replacement therapy (RRT).

## Methods

### Study population

In this retrospective observational cohort study, we screened all adult patients who were admitted to the intensive care unit (ICU) at Mayo Clinic Rochester between January 2006 and December 2016, who required vasopressor infusion therapy (i.e., norepinephrine, vasopressin,

epinephrine, phenylephrine, and dopamine). We excluded patients without research authorization, those with known pregnancy or end-stage renal disease, renal transplantation recipients, and individuals with the initiation of RRT before diuretic administration. Patients who received vasopressors prior to admission to the index ICU admission, who were in the ICU for >14 days prior to the vasopressor initiation, individuals with previous ICU admissions within 14 days or during the same hospitalization and patients with unknown vasopressor concentrations were also excluded. We also excluded patients with mechanical circulatory support, patients who were admitted to the cardiovascular surgery ICU or cardiac ICU as diuretic use of patients with cardiogenic shock is indicated early in the course of shock management. Patients who received two different inotropic agents on the first day of diuretic administration received a diuretic other than furosemide or bumetanide, had missing urine output data, who were on vasopressors for <6 hours, or received non-simultaneous vasopressors and diuretics were also excluded. This study was approved by the Mayo Clinic Institutional Review Board, and all activities were carried out in accordance with the modified Declaration of Helsinki. Informed consent was waived due to the minimal risk of the study.

## Data abstraction and definitions

The information included in this study was electronically abstracted. The Multidisciplinary Epidemiology and Translational Research in Intensive Care (METRIC) ICU DataMart, a near real-time the relational database was used to pull most data, and validated search queries were used when available [20]. All the data, including demographics, comorbidities, baseline characteristics, lab values, medication administration timing and dosing, laboratory values, and outcomes of interest such as acute kidney injury and initiation of RRT were abstracted electronically using validated digital algorithms when available. We collected hourly vasopressor doses and urinary outputs for ±6 hours of the first dose of diuretic administration. Time zero defined as the time of diuretic dose for the intervention group and the time vasopressor initiation for the control group.

We reported the cumulative vasopressor dose in norepinephrine equivalents using the conversion table included in angiotensin II for treatment in vasodilatory shock trial [21]. We converted all doses of loop diuretics to furosemide equivalents using published conversion tables in the cardiology literature [22]. Acute kidney injury (AKI) stage III was defined based on both serum creatinine levels and hourly urine output using the Kidney Disease Improving Global Outcomes criteria (KDIGO) criteria [23]. Estimated glomerular filtration rate (eGFR) was calculated using the Chronic Kidney Disease Epidemiology Collaboration (CKD-EPI) formula [24]. We defined oliguria as a urinary output of <0.5 ml/kg/hr averaged over 6 hours. We screened for oliguria in the 6 hours preceding the administration of the first diuretic dose. For those who did not receive diuretics, we assessed the presence of oliguria in the first 6 hours of vasopressor administration. Advanced oxygen therapy was defined as the use of mechanical ventilation, noninvasive ventilation, or high-flow nasal cannula.

## Statistical methods

Descriptive statistics are presented as the median and interquartile range (IQR) for continuous or interval variables, counts, and percentages for categorical variables. Prior to propensity matching, Wilcoxon-Mann-Whitney or Chi-Square tests were used to assess for differences between groups.

The effect of diuretic administration on urine output and vasopressor administration was assessed using linear mixed-effects models. The number of hours relative to the time of diuretic administration, the presence of oliguria, and administration of diuretic were included as covariates. All vasopressors were converted to equivalent micrograms of norepinephrine

per kilogram of patient weight. Both urine output and amount of vasopressor were transformed due to observed skew prior to the modeling. Model fit and choice of covariance matrix was assessed using Bayes information criterion (BIC). To account for potential variability over the period of diuretic administration, in addition to a simple diuretic by time interaction, models using piecewise linear sections over the diuretic period were examined. The optimal number and location of cut-points were selected based on the model fit. A single cut-point between two and three hours post-administration was chosen for the urine output models, and no cut-points were selected for the vasopressor administration model. A first-order autoregressive (AR (.1)) covariance structure was used in the final models.

Vasopressor alone group and diuretic/vasopressors treatment group were matched 1:1 using a greedy algorithm on the following variables [25]: presence of oliguria, gender, do not resuscitate/do not intubate (DNR/DNI) order before the inclusion, use of advanced oxygen therapies prior to inclusion, lowest lactate and partial pressure of oxygen to fraction of inspired oxygen ratio (P/F), age, eGFR, baseline SOFA score, and average urine output Caliper widths used for matches for other variables were 0.1 mL/hour/kg of body weight for urine output, 10 ml/min/1.73m$^2$ for GFR, 1 for SOFA score, 10 years for age, 1 for minimum lactate, and 100 for P/F ratio. After matching, Wilcoxon signed-rank, McNemar's Test, or Bowker's test were used to assess for residual differences between matched cases and controls. Statistical significance was treated as a p-value <0.05. Statistical analyses were performed using SAS (version 9.4; Cary, NC).

## Results

Among 168,833 screened patients, 150,856 were adults. Among those, 39,474 received vasopressors and 32,727 patients were excluded by a-priori exclusion criteria. Our final cohort included 929 cases who received concomitant diuretics and vasopressors and 5,818 potential controls (**Fig 1**). The characteristics of the population are included in **Table 1**. The median age of the vasopressor + diuretic group was 68 years (58, 78). Most patients were the White race (855, 92%), and 400 (43%) were females. In this group 549 (59%) were oliguric, and 764 (82%) were on the invasive mechanical ventilator before administration of the first diuretic dose. The median dose of vasopressor administered on the first day, represented in norepinephrine equivalents, was 39.3 mcg (IQR: 11.2, 90.0).

### Use of diuretics concomitant with vasopressors

The use of diuretics concomitant with vasopressors increased over time; in 2016, 16.43% of patients in the ICU received diuretics concomitantly with vasopressors compared to 11.4% in 2011 and 10.7% in 2006 (**Fig 2**).

### Effect of diuretics on urine output and vasopressor dose

The average hourly urine output during the first six hours following time zero in comparison with average hourly urine output during the six hours prior to time zero was significantly higher in diuretic group in comparison with patients who did not receive diuretics [81 (95% CI 73–89) ml/h vs. 42 (95% CI 39–45) ml/h, respectively; p<0.001] (**Fig 3A**). Likewise, the total urine output during the first six hours following time zero in comparison with total urine output during the six hours prior to time zero was significantly higher in diuretic group in comparison with patients who did not receive diuretics [485 (95% CI 440–531) ml vs. 251 (95% CI 233–269) ml, respectively; p<0.001] (**Table 1**, **S1A Fig.** and **S1B Fig.**). Among patients who received diuretics, the amount of vasopressor administered did not change from

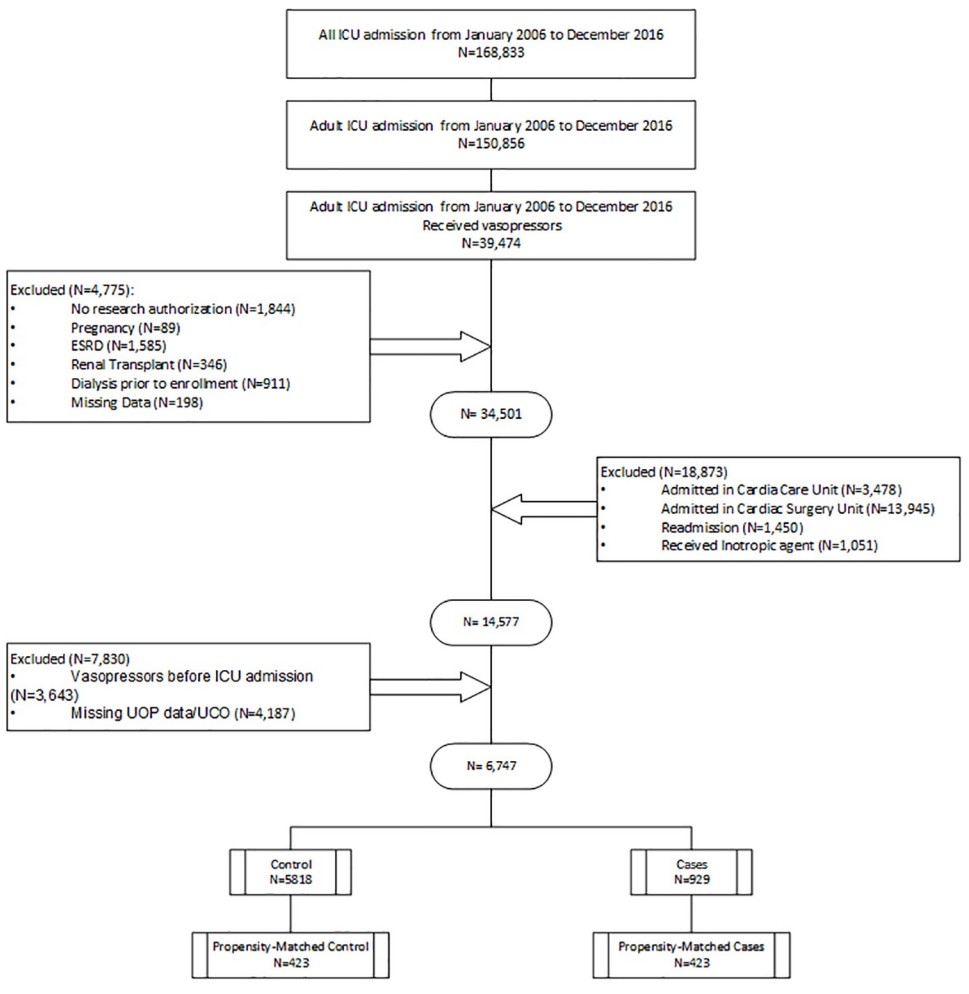

**Fig 1. Patient enrollment flow chart.**

6 hours prior to 6 hours after time zero (mean difference of norepinephrine equivalent dose of -0.06 with 95% CI -0.3 to 0.2 mg/day, p<0.001) (**Fig 3B**).

## Outcome measures: Acute kidney injury and initiation of renal replacement therapies

Prior to propensity matching, the median (IQR) of total urine output on the control group during the six hours after time zero (on vasopressors without diuretics) was 391 (186, 811) ml, while the median (IQR) urine output of the diuretic group during the 6 hours before time zero (on vasopressor before receiving diuretics) was 252 (133, 482) ml (P<0.001). In addition, using the competing risk model after accounting for the mortality, the sub-hazard risk ratio of AKI in the diuretic group in comparison with the control group was 1.65 (95% CI: 1.48–1.82, p < .001). After dividing the entire cohort based on the AKI status, defined as an increase in baseline serum creatinine by 1.5 times, as expected, the 90-day mortality of patients with AKI was significantly higher than those without AKI (OR 2.4, 95% CI 2.17, 2.68; P<0.001). Among those who did not receive diuretics, 415 (7%) and 504 (9%) received RRT in the first seven days and during the hospital admission, respectively. For the diuretic group, the number of patients who received RRT in the first seven days and during the hospital admission were 118

**Table 1. Characteristics and outcomes of patients who received diuretic therapy and those who did not before matching.**

| Variable | No diuretics N = 5818 | Diuretic N = 929 | Total = N 6747 | P-value |
|---|---|---|---|---|
| Age; Median (IQR) | 65 (54, 74) | 68 (57, 78) | 65 (55, 75) | < .001[1] |
| Gender, F, n (%) | 2526 (43%) | 400 (43%) | 2926 (43%) | .8[2] |
| CKD-EPI eGFR; Median (IQR) | 78 (57, 96) | 74 (53, 94) | 78 (57, 96) | < .001[1] |
| Oliguria, n (%) | 1430 (25%) | 465 (50%) | 1895 (29%) | < .001[2] |
| Advanced oxygen therapy, n (%) | 1906 (33%) | 718 (77%) | 2624 (39%) | < .001[2] |
| Min Lactate on inclusion | 1.5 (1.3, 1.7) | 1.5 (1.2, 1.8) | 1.5 (1.3, 1.7) | .9[1] |
| Minimum P/F ratio; Median (IQR) | 191 (165, 223) | 173 (144, 200) | 191 (162, 219) | < .001[1] |
| SOFA Score; Median (IQR) | 7 (4, 10) | 9 (7, 11) | 8 (5, 11) | < .001[1] |
| APACHE III Score; Median (IQR) | 49 (36, 64) | 53 (39, 68) | 49 (36, 65) | < .001[1] |
| Total Norepinephrine equivalent dose mcg/kg; Median (IQR) | 53 (20, 131) | 39 (11, 90) | 51 (19, 125) | < .001[1] |
| Day 1 Fluid Balance ml, (IQR) | 2790 (1138, 5410) | -116 (-1551, 1164) | 2361 (654, 4944) | < .001[1] |
| Day 1 Urine output ml, (IQR) | 1532 (848, 2667) | 2532 (1172, 3892) | 1616 (864, 2863) | < .001 |
| Day 2 Fluid Balance ml, (IQR) | 329 (-451, 1432) | -208 (-1481, 682) | 247 (-603, 1335) | < .001[1] |
| Day 2 Urine output ml, (IQR) | 1250 (630, 2277) | 2054 (913, 3448) | 1616 (864, 2863) | < .001 |
| Day 3 Fluid Balance ml, (IQR) | -43 (-981, 788) | -249 (-1456, 579) | -72 (-1062, 761) | < .001[1] |
| Day 3 Urine output ml, (IQR) | 1517 (663, 2772) | 2089 (962, 3346) | 1616 (864, 2863) | < .001 |
| Total UOP 6 hours before time zero, ml, (IQR) | 222 (44, 499) | 252 (133, 481) | 226 (57, 497) | 0.6 |
| Total UOP 6 hours after time zero, ml, (IQR) | 390 (185, 811) | 710 (309, 1290) | 422 (196, 874) | <0.001 |
| Average hourly UOP 6 hours before time zero, ml, (IQR) | 37 (7, 83) | 42 (22, 80) | 38 (10, 83) | 0.6 |
| Average hourly UOP 6 hours after time zero, ml, (IQR) | 65 (31, 135) | 118 (52, 215) | 70 (33, 146) | <0.001 |
| Urine output Hour 2, ml, (IQR) | 40 (0, 122) | 112 (23, 319) | 46 (2, 147) | < .001 |
| Urine output Hour 3, ml, (IQR) | 47 (4, 132) | 101 (20, 244) | 52 (6, 147) | < .001 |
| Urine output Hour 4, ml, (IQR) | 48 (6, 124) | 81 (16, 203) | 51 (7, 132) | < .001 |
| Urine output Hour 5, ml, (IQR) | 46 (6, 121) | 82 (16, 195) | 49 (7, 133) | < .001 |
| Urine output Hour 6, ml, (IQR) | 46 (4, 111) | 75 (17, 172) | 49 (6, 121) | < .001 |
| Volume overload at time zero (>5% weight gain since admission), n (%) | 454 (8%) | 354 (38%) | 808 (12%) | < .001 |
| Volume overload at time zero (>10% weight gain since admission), n (%) | 203 (4%) | 201 (22%) | 404 (6%) | < .001 |
| AKI Stage III Day 0 to 7 n (%) | 941 (16%) | 268 (29%) | 1209 (18%) | < .001[2] |
| RRT Day 0 to 7 n (%) | 415 (7%) | 118 (13%) | 533 (8%) | < .001[2] |

[1]Wilcoxon rank sum p-value

[2]Chi-Square p-value

(13%), and 151 (16%) received, respectively, which was significantly higher than the control group (p<0.001).

After propensity matching, the total number of matched pairs was N = 423 (N = 846 subjects). The rate of stage III AKI within 7 days of study inclusion was similar between patients who did not receive diuretics and those who received diuretics concomitantly with vasopressors (66 (15.6%) and 83 (19.6%), respectively p = 0.1). The rate of RRT within 7 days of inclusion was also similar between both groups (34 (8.0%) and 37 (8.7%), respectively p = 0.7) (Table 2). ICU and 90-day mortality rates in patients who did not receive diuretics were less than those who did receive diuretics concomitantly with vasopressors (60 (14%) and 92 (22%), respectively; p = .003 for ICU mortality and) 156 (37%) and 191 (45%); p = .009 for 90-day mortality.

## Discussion

Our study provides a unique perspective to the dilemma of when to initiate volume removal strategy following shock resuscitation among volume overloaded patients. To our knowledge,

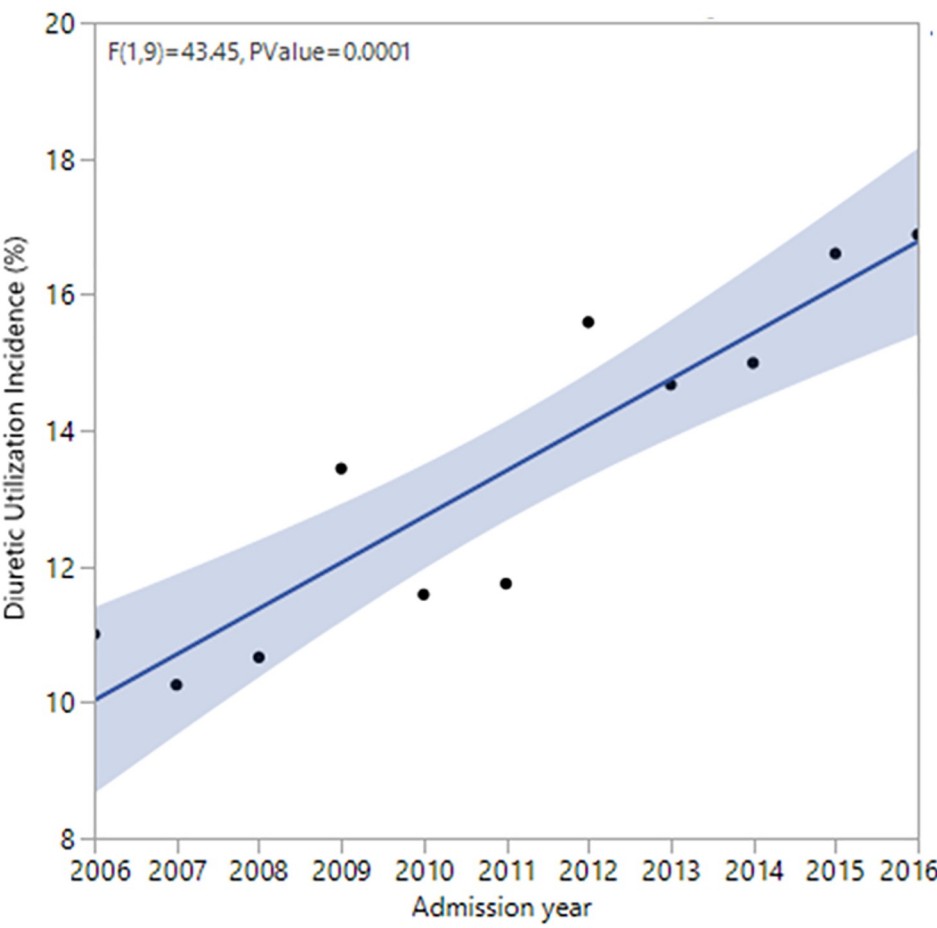

**Fig 2. Temporal trends of diuretic use among ICU adult patients who are on vasoactive agents.** The proportion of patients who received diuretics while on vasopressors was 11% in 2006, which increased to 17% in 2016 with an odds ratio of 1.06 per year of the study; P < .001.

this is the first study to address the use of loop diuretics in patients with the ongoing vasopressor infusion. This is an increasing practice in the management of critically ill patients, as observed in our study. We showed that diuretics augmented urine output despite vasopressor use. This can be very beneficial in achieving early volume control among volume overloaded patients in shock. We noted the vasopressor dose in the 6 hours preceding diuretic administration remained stable. After matching for baseline characteristics and factors that potentially affect the clinicians' decision-making processes in the administration of diuretics (i.e., oliguria, hypoxemia, use of advanced oxygenation therapies, DNR/DNI status), we found that patients on vasopressors who received diuretics are not at increased risk of developing AKI or need for RRT.

While the resuscitation of patients with shock state (particularly septic shock) is well-strategized and follows well accepted and published guidelines, de-escalation of resuscitated patients does not follow the same level of scrutiny and clarity. Questions like the time of initiation of volume removal, the rate of which volume could be removed with diuretics or renal replacement therapy without compromising hemodynamic state or clinical outcomes, the primary targets of volume removal (e.g., achieving admission weight; improvement in edema, laboratory variables, and organ function; reaching estimated dry weight), and safety of current strategies are yet to be answered.

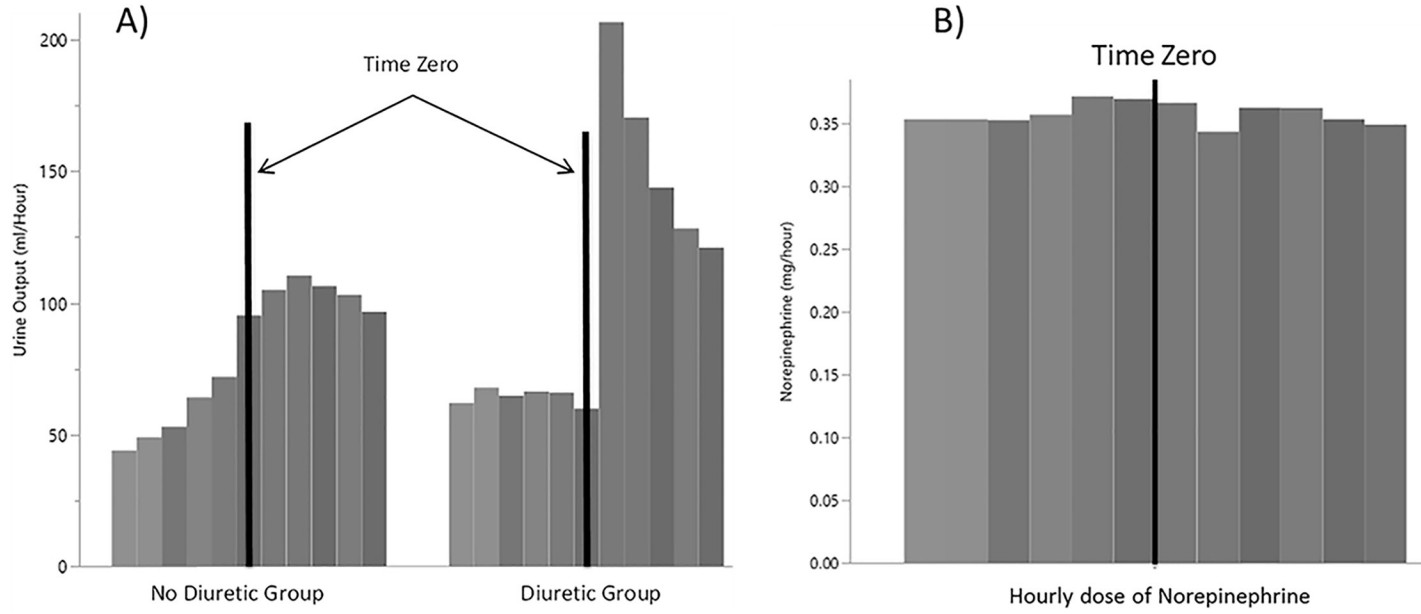

**Fig 3.** Effect of diuretics on A) urine output, and B) need for an increase in vasopressor doses.

It is clear that volume overload is associated with worse outcomes based on several observational studies [7, 26, 27]. Among volume overloaded patients, the extent of excess volume and also the length of time remaining volume overloaded have been found to be associated with higher mortality and morbidity rates. The strategies that are used among volume overloaded patients to remove volume include the use of diuretics, particularly loop diuretics, and

**Table 2. Characteristic and outcomes of patients with and without administration of diuretic therapy while on vasopressors after propensity matching.**

| | No diuretics N = 423 | Diuretic N = 423 | Total N = 846 | P-value |
|---|---|---|---|---|
| **Age–median (IQR)** | 69 (61, 78) | 69 (61, 77) | 69 (61, 77) | .1[1] |
| **Gender, F, n (%)** | 165 (39%) | 165 (39%) | 330 (39%) | * |
| **CKD-EPI eGFR Median (IQR)** | 70 (56, 92) | 73 (56, 93) | 71 (56, 92) | .6[1] |
| **Oliguria, n (%)** | 214 (51%) | 214 (51%) | 428 (51%) | * |
| **Advanced oxygenation therapies, n (%)** | 294 (70%) | 294 (70%) | 588 (70%) | * |
| **Min Lactate on inclusion** | 1.5 (1.2, 1.5) | 1.5 (1.2, 1.5) | 1.5 (1.2, 1.5) | .9[1] |
| **Minimum P/F ratio; Median (IQR)** | 191 (140, 191) | 173 (148, 178) | 173 (143, 191) | .06[1] |
| **SOFA Score Median (IQR)** | 8 (6, 10) | 8.0 (6, 10) | 8 (6, 10) | .5[1] |
| **APACHE III Score; Median (IQR)** | 53 (41, 66) | 54 (39, 68) | 53 (40, 67) | .9[1] |
| **Total Norepinephrine equivalent dose mcg/kg; Median (IQR) Day 1** | 61 (25, 138) | 37 (10, 75) | 47 (18, 111) | < .00[1] |
| **Day 1 fluid balance ml** | 2703 (1302, 5028) | -172 (-1609, 1139) | 1215 (-495, 3149) | < .001[1] |
| **Day 2 fluid balance ml** | 480 (-461, 1536) | -84 (-1281, 797) | 182 (-924, 1199) | < .001[1] |
| **Day 3 fluid balance ml** | -116 (-1036, 898) | -250 (-1410, 623) | -198 (-1232, 700) | .03[1] |
| **AKI Stage III Day 0 to 7 n (%)** | 66 (16%) | 83 (20%) | 149 (18%) | .1[2] |
| **RRT Day 0 to 7 n (%)** | 34 (8%) | 37 (9%) | 71 (9%) | .7[2] |

1 = Wilcoxon signed-rank test

2 = McNemar's Test

*McNemar's Test not calculated in cases that no discordant pairs were present

ultrafiltration. While diuretics could be used for patients that are not diuretic resistant and have reasonable kidney function, there are several safety concerns (e.g., decline in effective blood volume, hypotension, electrolyte imbalances, etc.) that limit their utilization, particularly in patients who are receiving vasopressors. Jeopardizing hemodynamic state, electrolyte, and acid-base imbalances, and rapid decrease in the effective blood volume are among concerns that clinicians may have when using diuretic agents [15, 16, 26].

The utilization of diuretic agents has changed over the course of the past few decades [28]. Trends of loop diuretics utilization for volume removal depend on the underlying reason for ICU admission. For example, in a very recent study, the use of diuretics was most common among patients after cardiac surgery followed by cardiac ICU patients followed by other medical, surgical, trauma patients. While patients with chronic kidney disease stage 5 used the least amount of diuretic, its utilization was rather common among other groups of patients with chronic kidney disease. Obviously, patients with acute decompensated heart failure and those on mechanical ventilation regardless of the type of ICU received the highest amount of loop diuretics [28]. Also, it seems patients with higher body mass index receive diuretics more frequently [29]. Interestingly, based on a large scale study, patients with chronic heart failure who depend on the maintenance of their volume balance for a better quality of life used a lower amount of loop diuretics [30]. The information regarding the temporal trends of diuretic utilization among patients which septic shock is scarce. We were able to demonstrate that the use of loop diuretics in patients who receive vasopressors is on an upward trajectory. These changes are most likely due to emerging knowledge regarding the detrimental impacts of volume overload potential benefits of its resolution.

While the frequency of AKI in the propensity-matched groups was not statistically different between the two groups, the observed differences in the entire cohort (before propensity matching) could potentially own clinical relevance (4% absolute and 25% relative increase risk when diuretics were used). Therefore, it is important that this notion to be validated in larger and prospective cohort studies. We noted that patients who received diuretics while on vasopressors had a higher rate of mortality in ICU and 90-days follow up. As this is a retrospective study, unknown confounding factors could explain this observation. During decision-making processes, those patients that receive diuretics are generally individuals who have more severe volume overload or oliguria when compared with those who do not receive diuretics. Despite our best efforts to adjust for the known factors, the differences in decision-making processes by the clinicians at the bedside could explain our observation regarding the mortality rates. A recent study by Shen and colleagues demonstrated improved mortality when diuretics were initiated early in the course of ICU admission. However, they did not explicitly specify whether patients were on vasopressors at the time of diuretic initiation [31]. Furthermore, they included patients on inotropes like dobutamine and did not specifically exclude patients with cardiogenic shock. Early administration of diuretics for volume removal could be part of the treatment of cardiogenic shock, and the inclusion of such patients in the cohort might have affected the outcomes in the mentioned study.

Our study has several limitations. Similar to other historical cohort studies, our investigation is subject to sampling bias. One of these potential confounders is that we could not rule out the presence of cardiogenic shock among those who were admitted in the medical ICU mainly due to missing data related to cardiac function among patients that were thought to have septic shock on admission. Another limitation of the retrospective studies is that there is no placebo for the control group. Therefore, time zero for the control group could have been different in comparison with the intervention group. In addition, as this is a single-center study, the results may not be generalizable to the other populations despite previously reported similarities in the rate of death with the national levels [32]. In addition, we are not able to

ascertain the causal relationship, and our findings are mainly associations. We did not have data on the presence of underlying cardiac dysfunction in the studied population, which can influence the decision to provide diuretics and affect outcomes in noncardiogenic shock patients. We are also limited in assessing the exact timing of diuretic administration in relation to the shock state. The timing of initiation of volume removal, early in the resuscitative phase of shock state or later in the resolution phase, possibly could affect outcomes even when patients are still receiving vasopressors. The retrospective nature of the study prohibits incorporating a clear clinical context in the decision to administer the diuretic therapy.

## Conclusion

Our study showed that the use of diuretics concomitantly with vasopressors is an increasing trend. While the use of diuretics is effective in augmentation of urine output, it does not increase the vasopressor dose. The patients who received diuretics did not have an increased risk of developing AKI or requiring more RRTs. However, they had a higher rate of ICU and 90-day mortality. Further studies and prospective trials are required to assess the role of loop diuretics in volume management of critically ill patients in shock.

## Supporting information

**S1 Fig.** Comparison between total urine output in 6 hours before and after time zero in A) control group, B) diuretic group.
(DOCX)

## Acknowledgments

We would like to acknowledge Shalini Donithi for her contributions to our project.

## Author Contributions

**Conceptualization:** Ghassan Bandak, Ognjen Gajic, Kianoush Kashani.

**Data curation:** Ghassan Bandak, Ankit Sakhuja, Nicole M. Andrijasevic.

**Formal analysis:** Tina M. Gunderson.

**Funding acquisition:** Ghassan Bandak, Kianoush Kashani.

**Investigation:** Ankit Sakhuja, Kianoush Kashani.

**Methodology:** Ognjen Gajic, Kianoush Kashani.

**Supervision:** Ognjen Gajic, Kianoush Kashani.

**Validation:** Kianoush Kashani.

**Visualization:** Kianoush Kashani.

**Writing – original draft:** Ghassan Bandak, Ankit Sakhuja, Kianoush Kashani.

**Writing – review & editing:** Ognjen Gajic, Kianoush Kashani.

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
