## [Decision Letter · Decision Letter 0]

12 Sep 2019

PONE-D-19-22578

Use of diuretics in shock: Trends and impacts

PLOS ONE

Dear Dr. Kashani,

Thank you for submitting your manuscript to PLOS ONE. After careful consideration, we feel that it has merit but does not fully meet PLOS ONE’s publication criteria as it currently stands. Therefore, we invite you to submit a revised version of the manuscript that addresses the points raised during the review process.

We would appreciate receiving your revised manuscript by Oct 27 2019 11:59PM. To enhance the reproducibility of your results, we recommend that if applicable you deposit your laboratory protocols in protocols.io, where a protocol can be assigned its own identifier (DOI) such that it can be cited independently in the future. For instructions see: http://journals.plos.org/plosone/s/submission-guidelines#loc-laboratory-protocols

We look forward to receiving your revised manuscript.

Kind regards,

Chiara Lazzeri

Academic Editor

PLOS ONE

Journal Requirements:

2. In the ethics statement in the manuscript and in the online submission form, please provide additional information about the patient records used in your retrospective study. Specifically, please ensure that you have discussed whether all data/tissue samples  were fully anonymized before you accessed them and/or whether the IRB or ethics committee waived the requirement for informed consent. If patients provided informed written consent to have data from their medical records used in research, please include this information.

Reviewers' comments:

Reviewer's Responses to Questions

**Comments to the Author**

1. Is the manuscript technically sound, and do the data support the conclusions?

Reviewer #1: Yes

Reviewer #2: Partly

2. Has the statistical analysis been performed appropriately and rigorously? 

Reviewer #1: Yes

Reviewer #2: No

3. Have the authors made all data underlying the findings in their manuscript fully available?

Reviewer #1: Yes

Reviewer #2: No

4. Is the manuscript presented in an intelligible fashion and written in standard English?

Reviewer #1: Yes

Reviewer #2: Yes

5. Review Comments to the Author

Reviewer #1: In this retrospective propensity matched study, the authors assessed the effect of loop diuretics on a subset of ICU patients in shock: namely those that received vasopressors. This interesting manuscript adds to the emerging literature in the field of critically ill volume overloaded patients and the potential risk and benefit of timely addressing the aforementioned volume overload.

The findings of this retrospective study could be summarized as follows:

1. A temporal trend was identified, as there was increased utilization of diuretics in shock patients that required pressors over the course of the decade that the study focused on (2006-2016)

2. The use of diuretics in these patients augmented their urine output (6hs post administration)

3. The use of diuretics did not result in increased AKI, need for RRT within the first week after their administration

4. Increased ICU mortality and 90 day mortality in patients that received diuretics compared to those that did not, even though residual bias might be driving those differences as patients receiving diuretics on top of vasopressors are commonly more critically ill.

Overall it is an interesting and well thought and constructed study that could be published in Plos One after certain points are addressed.

Major points

1. According to the results of the study, the use of diuretics did not lead to a significant increase in AKI but did result in increased ICU and 90-day mortality. Even if this difference could be attributed to residual confounding (cannot be sure at this point), it should be mentioned clearly in the abstract and/or the conclusion section.

2. Regarding the p-value of the AKI (p=0.11). The detected difference is clinically significant enough (19.6% vs 15.6% = 4% absolute or 25% relative) to warrant more thorough investigation or at least commentary, even if the p value is >0.05. If the matching for example was 1:2 instead of 1:1 this p value could have been leaning more towards significance. Authors should comment on the possible side effects of the diuretic use in the setting of the above and the increased mortality, for good measure.

There are also several limitations to this study. The authors did a good job of addressing those limitations in the limitations section. However, there are some minor additional points that should be clarified

1. Referral bias, as Mayo Clinic in Rochester, MN is a prominent referral center and the pool of patients might be meaningfully different than other populations across the globe, or even within the USA. As a result, extrapolation of those results should be made with caution, at minimum. I was pleased to see that the authors clearly stated that their results did not infer causality but a comment concerning possible referral bias would be welcome.

2. Although the use of English is generally appropriate and the points that the authors are trying to make are for the most part clearly stated, the use of English and the syntax is confusing in some occasions. Also, several typos were identified. Some examples of those mistakes that affect the flow of the manuscript are highlighted below. The authors are advised to reassess the language and syntax of their manuscript. Most of those mistakes are minor however and could easily be corrected.

“Among volume overloaded patients, the extent of excess volume overload and also the length of time remaining volume overloaded our variables that have been found to be associated with higher mortality and morbidity rates”

– “our variables” seems to be redundant

“there are several safety concerns that avoid their utilization”

-unclear

“Obviously, patients with heart failure and those on mechanical ventilation regardless of the type of ICU they received the highest amount of loop diuretics. [24] ….. Based on a large scale study, patients with heart failure who depend on the maintenance of their volume balance for a better quality of life used a lower amount of loop diuretics. [26]”

- contradictory? or are the authors referring to a subset of HF patients? Unclear without referring to the cited article.

Plos One specific comments:

As far as the reporting is concerned the following assessment is based on the latest STORBE statement.

Title/abstract: a balanced summary of what was done and what was found is provided in the abstract, with the exception of the mortality (see above)

Intro: the rationale behind the design of this study is successfully outlined in the intro

Methods: The authors did an excellent job for the most part of this section. Specifically, the study design is stated early on and the exclusion criteria are thoroughly outlined. The matching process is clearly described in the manuscript. Variables and data sources were also adequately identified in the Methods section. Additionally, the authors successfully addressed the potential variability of the period of diuretic administration. Also, the process of reaching the final study size is well documented in Fig 1. (flow diagr)

Results: # of participants at each level of the study was reported and clearly described with the help of a diagram (fig 1). Baseline characteristics of participants were also clearly provided in table 1. Also, results were properly outlined in the text and in tables 2 and 3

Discussion: the authors did summarize and put their results in context taking into account the available literature on the matter. They did a good job with that especially given the scarcity of quality data around diuretic use in critically ill (and volume overloaded) patients. Also, in this section the authors do a great job at reporting the increased ICU and 90 day mortality in patients that received diuretics and the potential reason behind that difference (patients receiving diuretics might be “sicker” than the ones not receiving). However, this difference should be clearly outlined in other sections of the manuscript as well (eg conclusions, abstract).

Also, a clear comment about the generalizability of the results could be incorporated at this section, given the possible referral bias of the study and its retrospective design that can only identify associations and not causality (authors did a good job of stating that)

Conclusions: the conclusions described in this section appropriately summarize the results of the study, with the exception of the statistically significant mortality increase (ICU mortality and 90 day mortality).

Lastly, potential funding that any of the investigators received and is pertinent to this study could be disclosed.

Overall this is an interesting and for the most part well-constructed manuscript that could add to the emerging field of the management of critically ill volume overloaded patients. My overall recommendation would be for this manuscript to be accepted for publication in Plos One after the above points are addressed.

Reviewer #2: Comments about the manuscript PONE-D-19-22578: “Use of diuretics in shock: Trends and impacts”.

This is a retrospective observational study about diuretics use during shock in critically ill patients. Some major concerns are noteworthy.

Title:

• Although PLOS One does not specify any title format, the present one sounds like a narrative review. Perhaps the authors might consider some usual format like “…retrospective cohort” or “…temporal trend and clinical impacts”.

Abstract:

• I suggest to clearly specify that the matched “control” cohort are patients also on vasopressors that did not receive diuretics.

• I would consider presenting the mixed model results in a different way. To report the standard error (also the authors should specify abbreviations when first cited) might not be “clinically informative”. Perhaps just describe the mean values before and after diuretics, or marginal means?

Introduction

• The first phrase of the final paragraph should be revised, since the authors did not evaluate the timing of diuretics initiation.

Methods

• What was the vasopressor use definition for the study? Any vasopressor? Any dose?

• One major point is the length of stay of patients prior to study inclusion. As far as I could understand, the authors excluded patients if they stayed in the ICU form “>14 days prior to the vasopressor initiation”. But usually length of stay correlates with severity and organ dysfunction. Patients who started vasopressors and/or diuretics later (e.g. after one week) might be systematically different from those who started these drugs earlier. This probably need to be taken into account on the statistical analysis, and at least be presented (i.e. length of stay in the ICU in the two matched groups).

• The authors did not include patients from cardiac ICU due to “cardiogenic shock”, but they should present the reasons for vasopressor use and/or causes of shock. Many patients with cardiogenic shock might have been included in the study, or the policy of your institution is to admit these patients only in cardiac ICUs?

• The authors excluded patients if they did not receive “vasopressors and diuretics during the first day of screening”, but it was not clearly stated which the first day of screening was. The first day in the ICU? The first day of concurrent use of vasopressor and diuretic? The first day of vasopressor use? Please clarify.

• The authors collected data in the diuretic group in a 6-hour window before and after the first dose of diuretic. In those without diuretic, they collected data in a similar time window, but before and after initiation of vasopressors. As stated before, these two time windows are probably different in the clinical course of the patients. It might be expected that diuretics are administrated later (i.e. after a few hours or even days of the beginning of vasopressors). So the two time windows might not be the same and this fact should influence the results. The authors should clearly describe these two time windows (i.e. at least the mean length of stay in the ICU before these windows).

• The authors applied the KDIGO criteria for AKI diagnosis. However, the use of diuretics might influence urine output. So the authors should apply only the creatinine criteria (or RRT start).

• The oliguria screening might suffer from the same time window bias as described before.

• For the mixed-model, the urine output and vasopressor rate were the “dependent variables”. But were they modelled as the mean before and after diuretics or the amount (volume of urine or mass of vasopressor) per kg per hour hourly in a segmented regression?

• As far as I could understand the authors did not take into account in the mixed model the severity of illness (such as SOFA at the day of study). And they did not model the use of diuretic and the dose of vasopressor in the same model (they stated that the covariates were the hours relative to diuretic administration, oliguria and the use of diuretics). I believe that the amount of vasopressors should be equally relevant for the diuretic response and diuretic effect on arterial pressure. Please clarify.

• The authors should provide some graph to show that the cut-off between two to three hours is adequate for the segmented regression (e.g. with LOESS for the graph).

• Perhaps the behavior of mean arterial pressure would be more informative and clinically relevant than the vasopressors dose (and might not have a skewed distribution).

Results

• Results in table 1 are similar to data in table 2. Please review.

• The legend below table 1 is probably a note for the authors. Please review.

• Data of concomitant use of diuretics with vasopressors in figure 2 should be tested with some test for trend.

• A figure with the time evolution of urine output before and after diuretics would be welcomed. The same goes for vasopressors dose and mean arterial pressure.

• Table 2 does not have data for urine output hourly as stated in the manuscript. Please review.

• SOFA values in table 3 are wrong. Please review.

• Since mortality rates were significantly different between groups (diuretics vs no diuretics), the incidence of AKI should be evaluated with a competing risk model.

Discussion

• One of the limitations is the long study period. Clinical practice might have changed (e.g. the pattern of diuretics use as stated by the authors) during the study period and influence the results.

6. PLOS authors have the option to publish the peer review history of their article (what does this mean?). If published, this will include your full peer review and any attached files.

Reviewer #1: No

Reviewer #2: No

---

## [Author Response · Author response to Decision Letter 0]

3 Jan 2020

Response to reviewers: 

Reviewer #1: 

In this retrospective propensity matched study, the authors assessed the effect of loop diuretics on a subset of ICU patients in shock: namely those that received vasopressors. This interesting manuscript adds to the emerging literature in the field of critically ill volume overloaded patients and the potential risk and benefit of timely addressing the aforementioned volume overload.

The findings of this retrospective study could be summarized as follows:

1. A temporal trend was identified, as there was increased utilization of diuretics in shock patients that required pressors over the course of the decade that the study focused on (2006-2016)

2. The use of diuretics in these patients augmented their urine output (6hs post administration)

3. The use of diuretics did not result in increased AKI, need for RRT within the first week after their administration

4. Increased ICU mortality and 90 day mortality in patients that received diuretics compared to those that did not, even though residual bias might be driving those differences as patients receiving diuretics on top of vasopressors are commonly more critically ill.

Overall it is an interesting and well thought and constructed study that could be published in Plos One after certain points are addressed.

Authors’ Response: Thank you very much for the excellent summary and positive feedback.

Major points

1. According to the results of the study, the use of diuretics did not lead to a significant increase in AKI but did result in increased ICU and 90-day mortality. Even if this difference could be attributed to residual confounding (cannot be sure at this point), it should be mentioned clearly in the abstract and/or the conclusion section.

Authors’ Response: Thank you for this point. This is now mentioned in the abstract and conclusion section of the paper. 

2. Regarding the p-value of the AKI (p=0.11). The detected difference is clinically significant enough (19.6% vs 15.6% = 4% absolute or 25% relative) to warrant more thorough investigation or at least commentary, even if the p value is >0.05. If the matching for example was 1:2 instead of 1:1 this p value could have been leaning more towards significance. Authors should comment on the possible side effects of the diuretic use in the setting of the above and the increased mortality, for good measure.

Authors’ Response: Thank you for this interesting point. Type II error could always be a possibility in statistical analyses. However, in the statistical analysis plans for this study, we defined p-value <0.05 for statistical significance. Using this argument in the result section would be against our a-priori analytic plans. However, your important point is mentioned in the discussion section added to the recommendation for further analyses. 

“While the frequency of AKI was not statistically different between the two groups, the observed differences could be potentially own clinical relevance (4% absolute and 25% relative increase risk when diuretics were used). Therefore, it is important that this notion be validated in larger and prospective cohort studies.”

There are also several limitations to this study. The authors did a good job of addressing those limitations in the limitations section. However, there are some minor additional points that should be clarified

1. Referral bias, as Mayo Clinic in Rochester, MN is a prominent referral center and the pool of patients might be meaningfully different than other populations across the globe, or even within the USA. As a result, extrapolation of those results should be made with caution, at minimum. I was pleased to see that the authors clearly stated that their results did not infer causality but a comment concerning possible referral bias would be welcome.

Authors’ Response: We agree Mayo Clinic is considered a regional center with a large referral base. However, based on previous data, the severity of illness and mortality rate is not different from the general population at the US national levels.[1]

we also added: " In addition, as this is a single-center study, the results may not be generalizable to the other populations despite previously reported similarities in the rate of death with the national levels.”

2. Although the use of English is generally appropriate and the points that the authors are trying to make are for the most part clearly stated, the use of English and the syntax is confusing in some occasions. Also, several typos were identified. Some examples of those mistakes that affect the flow of the manuscript are highlighted below. The authors are advised to reassess the language and syntax of their manuscript. Most of those mistakes are minor however and could easily be corrected.

“Among volume overloaded patients, the extent of excess volume overload and also the length of time remaining volume overloaded our variables that have been found to be associated with higher mortality and morbidity rates”

– “our variables” seems to be redundant

Authors’ Response: We apologize for these mistakes. We thoroughly reviewed the script to correct all mistakes. This sentence was changed to: “Among volume overloaded patients, the extent of excess volume and also the length of time remaining volume overloaded have been found to be associated with higher mortality and morbidity rates.”

“there are several safety concerns that avoid their utilization”

-unclear

Authors’ Response: For further clarification this sentence was changed to: “While diuretics could be used for patients that are not diuretic resistant and have reasonable kidney function, there are several safety concerns (e.g., decline in effective blood volume, hypotension, electrolyte imbalances, etc.) that limit their utilization, particularly in patients who are receiving vasopressors.”

“Obviously, patients with heart failure and those on mechanical ventilation regardless of the type of ICU they received the highest amount of loop diuretics. [24] ….. Based on a large scale study, patients with heart failure who depend on the maintenance of their volume balance for a better quality of life used a lower amount of loop diuretics. [26]”

- contradictory? or are the authors referring to a subset of HF patients? Unclear without referring to the cited article.

Authors’ Response: we were trying to point out the lack of consistent literature, particularly in acute and chronic settings, and the need for more studies. for further clarification we changed these sentences as below:

“methods with acute decompensated heart failure and those on mechanical ventilation regardless of the type of ICU received the highest amount of loop diuretics.”

“This is while based on a large scale study, patients with chronic heart failure who depend on the maintenance of their volume balance for a better quality of life used a lower amount of loop diuretics.”

 

Plos One specific comments:

As far as the reporting is concerned the following assessment is based on the latest STORBE statement.

Title/abstract: a balanced summary of what was done and what was found is provided in the abstract, with the exception of the mortality (see above)

Authors’ Response: Thank you. Mortality information was added to the abstract.

Intro: the rationale behind the design of this study is successfully outlined in the intie

Methods: The authors did an excellent job for the most part of this section. Specifically, the study design is stated early on and the exclusion criteria are thoroughly outlined. The matching process is clearly described in the manuscript. Variables and data sources were also adequately identified in the Methods section. Additionally, the authors successfully addressed the potential variability of the period of diuretic administration. Also, the process of reaching the final study size is well documented in Fig 1. (flow diagr)

Results: # of participants at each level of the study was reported and clearly described with the help of a diagram (fig 1). Baseline characteristics of participants were also clearly provided in table 1. Also, results were properly outlined in the text and in tables 2 and 3

Discussion: the authors did summarize and put their results in context taking into account the available literature on the matter. They did a good job with that especially given the scarcity of quality data around diuretic use in critically ill (and volume overloaded) patients. Also, in this section the authors do a great job at reporting the increased ICU and 90 day mortality in patients that received diuretics and the potential reason behind that difference (patients receiving diuretics might be “sicker” than the ones not receiving). However, this difference should be clearly outlined in other sections of the manuscript as well (eg conclusions, abstract).

Authors’ Response: Thank you. We have had 90-day mortality information to the abstract and conclusions

Also, a clear comment about the generalizability of the results could be incorporated at this section, given the possible referral bias of the study and its retrospective design that can only identify associations and not causality (authors did a good job of stating that)

Authors’ Response: We agree Mayo Clinic is considered a regional center with a large referral base. However, based on previous data, the severity of illness and mortality rate is not different from the general population at the US national levels.[1]

we also added: " In addition, as this is a single-center study, the results may not be generalizable to the other populations despite previously reported similarities in the rate of death with the national levels.”

Conclusions: the conclusions described in this section appropriately summarize the results of the study, with the exception of the statistically significant mortality increase (ICU mortality and 90 day mortality).

Lastly, potential funding that any of the investigators received and is pertinent to this study could be disclosed.

Authors’ Response: Thank you for the thorough assessment review of our paper. 

Overall this is an interesting and for the most part well-constructed manuscript that could add to the emerging field of the management of critically ill volume overloaded patients. My overall recommendation would be for this manuscript to be accepted for publication in Plos One after the above points are addressed.

Authors’ Response: Thank you for the thorough assessment review of our paper. We appreciate encouraging comments.

 

Reviewer #2: Comments about the manuscript PONE-D-19-22578: “Use of diuretics in shock: Trends and impacts”.

This is a retrospective observational study about diuretics use during shock in critically ill patients. Some major concerns are noteworthy.

Title:

• Although PLOS One does not specify any title format, the present one sounds like a narrative review. Perhaps the authors might consider some usual format like “…retrospective cohort” or “…temporal trend and clinical impacts”.

Authors’ Response: We thank the reviewer for this point. We have changed the title to:

“Use of Diuretics in Shock: Temporal Trends and Clinical Impacts in a Propensity Matched Cohort Study”

Abstract:

• I suggest to clearly specify that the matched “control” cohort are patients also on vasopressors that did not receive diuretics.

Authors’ Response: Thank you for this point. We have clarified the abstract has recommended.

• I would consider presenting the mixed model results in a different way. To report the standard error (also the authors should specify abbreviations when first cited) might not be “clinically informative.” Perhaps just describe the mean values before and after diuretics, or marginal means?

Authors’ Response: This information is provided as requested by the reviewer.

“The average hourly urine output during the first six hours following time zero in comparison with average hourly urine output during the six hours prior to time zero was significantly higher in diuretic group in comparison with patients who did not receive diuretics [81 (95% CI 73-89) ml/h vs. 42 (95% CI 39-45) ml/h, respectively; p<0.001]. Hence, the total urine output during the first six hours following time zero in comparison with total urine output during the six hours prior to time zero was significantly higher in diuretic group in comparison with patients who did not receive diuretics [485 (95% CI 440-531) ml vs. 251 (95% CI 233-269) ml, respectively; p<0.001] (Table 2). Among patients who received diuretics, the amount of vasopressor administered did not change from 6 hours prior to 6 hours after time zero (mean difference of norepinephrine equivalent dose of (mg/day) -0.06 with 95% CI -0.3 to 0.2, p<0.001).”

Introduction

• The first phrase of the final paragraph should be revised, since the authors did not evaluate the timing of diuretics initiation.

Authors’ Response: Thank you for this comment. This phrase was changed to:

“We believe there is equipoise regarding the initiation of the fluid de-escalation phase of resuscitation while patients are still on vasopressors.”

Methods

• What was the vasopressor use definition for the study? Any vasopressor? Any dose?

Authors’ Response: Regarding the vasopressor type, we included all vasoactive medications, including and limited to norepinephrine, vasopressin, epinephrine, phenylephrine, dopamine. Then, we converted doses to norepinephrine equivalents using published tables (included in our methods sections). The number of different types of vasopressors used was not different between the 2 matched groups (P 0.57). We did not have any exclusion criteria based on dose, but excluded patients who received vasopressors for < 6 hours (included in our methods section).

For further clarification, we changed the first sentence of the method section to:

“In this retrospective observational cohort study, we screened all adult patients who were admitted to the intensive care unit (ICU) at Mayo Clinic Rochester between January 2006 and December 2016 who required vasopressor infusion therapy (i.e., norepinephrine, vasopressin, epinephrine, phenylephrine, dopamine).”

• One major point is the length of stay of patients prior to study inclusion. As far as I could understand, the authors excluded patients if they stayed in the ICU form “>14 days prior to the vasopressor initiation”. But usually length of stay correlates with severity and organ dysfunction. Patients who started vasopressors and/or diuretics later (e.g. after one week) might be systematically different from those who started these drugs earlier. This probably need to be taken into account on the statistical analysis, and at least be presented (i.e. length of stay in the ICU in the two matched groups).

Authors’ Response: Thank you for sharing this point. We did feel this could represent a limitation. However, we did match for organ dysfunction using SOFA scores, the use of advanced oxygenation devices, hypoxemia, and eGFR, which we felt could help mitigate this concern. 

In addition, we excluded these patients as these individuals most likely had entered the “Persistent Critical Illness” domain when their prognosis is not determined by the original reason for ICU admission any longer. Persistent critical illness is associated with poor outcomes and increased utilization of health-care resources, posing a substantial burden upon patients, families, and society on its own. Although the clinical presentation may vary, a predominant feature is the failure to wean from mechanical ventilation, accompanied by severe catabolism and intensive care unit (ICU)-acquired weakness [2]. The transition from acute to persistent critical illness occurs over days. The onset of persistent critical illness was recently defined as the time point at which admission diagnosis and illness severity no longer predict outcomes better than pre-existent patient characteristics. This generally occurs within one to three weeks after ICU admission, with the average lying around day 10 [3].

• The authors did not include patients from cardiac ICU due to “cardiogenic shock”, but they should present the reasons for vasopressor use and/or causes of shock. Many patients with cardiogenic shock might have been included in the study, or the policy of your institution is to admit these patients only in cardiac ICUs?

Authors’ Response: The reviewer is raising a very important point, which is one of the reasons for bias in retrospective studies. We tried to exclude cardiogenic shock patients by not including patients who were admitted to the cardiac ICU. Unfortunately, the volume of missing information to differentiate the septic shock from cardiogenic shock in the medical ICU is beyond a meaningful analysis. We have mentioned this in the limitation section. 

“One of these potential confounders is that we could not rule out the presence of cardiogenic shock among those who were admitted in the medical ICU mainly due to missing data related to cardiac function among patients that were thought to have septic shock on admission.”

• The authors excluded patients if they did not receive “vasopressors and diuretics during the first day of screening”, but it was not clearly stated which the first day of screening was. The first day in the ICU? The first day of concurrent use of vasopressor and diuretic? The first day of vasopressor use? Please clarify.

Authors’ Response: Thank you. Our statement was “Patients who received two different inotropic agents on the first day of diuretic administration received a diuretic other than furosemide or bumetanide, had missing urine output data, who were on vasopressors for <6 hours, or received non-concurrent vasopressors and diuretics during the first day of vasopressor initiation were also excluded.” 

The first day was defined as the day of vasopressor was initiated. What the statement says is to clarify that vasopressor and diuretics needed to be simultaneous. We have made the following edit to address this concern:

“Patients who received two different inotropic agents on the first day of diuretic administration received a diuretic other than furosemide or bumetanide, had missing urine output data, who were on vasopressors for <6 hours, or received nonconcurrent vasopressors and diuretics were also excluded.”

• The authors collected data in the diuretic group in a 6-hour window before and after the first dose of diuretic. In those without diuretic, they collected data in a similar time window, but before and after initiation of vasopressors. As stated before, these two time windows are probably different in the clinical course of the patients. It might be expected that diuretics are administrated later (i.e. after a few hours or even days of the beginning of vasopressors). So the two time windows might not be the same and this fact should influence the results. The authors should clearly describe these two time windows (i.e. at least the mean length of stay in the ICU before these windows).

Authors’ Response: Thank you for raising this important point. The time of initiation of vasopressors was the same between groups. As the control group did not receive diuretic, the time zero for them should have been decided a-priori. We were not able to decide on another time point for the control group. As this may also induce some bias, we have added this to the limitation section.

“Another limitation of the retrospective studies is that there is no placebo for the control group. Therefore, time zero for the control group could have been different in comparison with the intervention group.”

• The authors applied the KDIGO criteria for AKI diagnosis. However, the use of diuretics might influence urine output. So the authors should apply only the creatinine criteria (or RRT start).

Authors’ Response: Thank you. Although the reviewer is correct in spirit of the definition, the KDIGO criteria do not provide any provision for the definition of AKI based on urine output after receipt of diuretics differently from those who do not receive diuretics. Therefore, despite recognizing this deficit of definition, we believe our paper is still according to the current criteria for the definition of AKI. Indeed, arguably, those who remain oliguric despite diuretics may have suffered from more intense AKI.

• The oliguria screening might suffer from the same time window bias as described before.

Authors’ Response: Thank you for raising this important point. The time of initiation of vasopressors was the same between groups. As the control group did not receive diuretic, the time zero for them should have been decided a-priori. We were not able to decide on another time point for the control group. As this may also induce some bias, we have added this to the limitation section.

“Another limitation of the retrospective studies is that there is no placebo for the control group. Therefore, time zero for the control group could have been different in comparison with the intervention group.” 

• For the mixed-model, the urine output and vasopressor rate were the “dependent variables”. But were they modelled as the mean before and after diuretics or the amount (volume of urine or mass of vasopressor) per kg per hour hourly in a segmented regression?

Authors’ Response: Thank you. While this is a great suggestion, following several conversations among the authors, we felt this would be out of the scope of this study. Thank the reviewer for the suggestion. Please see the Kernel Smoothed fit model in Supplementary Figures 2A and 2B.

• As far as I could understand the authors did not take into account in the mixed model the severity of illness (such as SOFA at the day of study). And they did not model the use of diuretic and the dose of vasopressor in the same model (they stated that the covariates were the hours relative to diuretic administration, oliguria and the use of diuretics). I believe that the amount of vasopressors should be equally relevant for the diuretic response and diuretic effect on arterial pressure. Please clarify.

Authors’ Response: Thank you for the interesting comment. In the mixed-models, we did not include the variables that did not reach our a-priori BIC threshold. Unfortunately, the amount of vasopressors included a significantly large number of missing information, so it was not included in the model. In order to address the reviewer’s concern, we would like to remind the reviewer that 1:1 propensity match was done to allow the dose of vasopressors and severity of illness scores to be similar (Table 3)

• The authors should provide some graph to show that the cut-off between two to three hours is adequate for the segmented regression (e.g. with LOESS for the graph).

Authors’ Response: Thank you for this very interesting suggestion. We have added the LOESS (Kernel Smother) plots to the supplementary material.

• Perhaps the behavior of mean arterial pressure would be more informative and clinically relevant than the vasopressors dose (and might not have a skewed distribution).

Authors’ Response: Thank you for the interesting comment. All patients included in this study were on vasopressors. At our institution, we use vasopressors to meet specific mean arterial pressure goals, and thus the dose of medication used will be a more suitable indicator of hemodynamics since the goal of mean arterial pressure would be achieved with titration of vasopressors doses.

Results

• Results in table 1 are similar to data in table 2. Please review.

Authors’ Response: The reviewer is correct, as the information in Table 1 is included in the second column of Table 2. However, Table 1 also includes variables like race, which is not included in Table 2. While we are open to remove this table if the reviewer and editor recommend, we feel it still adds additional information.

• The legend below table 1 is probably a note for the authors. Please review.

Authors’ Response: The reviewer is correct. We removed the footnote, as suggested.

• Data of concomitant use of diuretics with vasopressors in figure 2 should be tested with some test for trend.

Authors’ Response: Thank you. The proportion of patients who received diuretics while on vasopressors was 11% in 2006, which increased to 17% in 2016 with an odds ratio of 1.06 per year of the study and P<.001. This was added to the footnote of the figure, as suggested.

• A figure with the time evolution of urine output before and after diuretics would be welcomed. The same goes for vasopressors dose and mean arterial pressure.

Authors’ Response: Thank you. Figures 3A and 3B were added, as suggested by the reviewer.

• Table 2 does not have data for urine output hourly as stated in the manuscript. Please review.

Authors’ Response: Thank you. This information is added to Table 2.

• SOFA values in table 3 are wrong. Please review.

Authors’ Response: Thank you. This was corrected.

• Since mortality rates were significantly different between groups (diuretics vs. no diuretics), the incidence of AKI should be evaluated with a competing risk model.

Authors’ Response: When we defined AKI based on an increase in baseline serum creatinine by 1.5 times, the sub-hazard ratio for any AKI, after considering death in ICU or within the first seven days after time zero as the competing outcome, was 1.65 (95% CI: 1.48-1.82, p<.001) in diuretic group compared to the controls. This was obviously expected as those who received diuretics while on vasopressors had significantly higher chances of being oliguric and, therefore, an increased risk of AKI.

When divided the whole cohort for AKI (defined as an increase in baseline serum creatinine by 1.5 times), while 90-day mortality of patients with AKI was significantly higher than those without AKI, the group who received diuretics regardless of AKI status, has higher mortality rate (P<0.001):

Patients without AKI:

Patients with AKI:

Study_Case 0 = no diuretics; Study_Case 1 = diuretics; 90 Day Mortality 0 = Alive; 90 Day Mortality 1 = Dead

This again could be potentially explained as it was in the first paragraph of this question. Those who received diuretics were more likely to be oliguric (in Supplementary Figure 1 the urine output of control group after time zero [i.e., on vasopressor and not on diuretic] is more than urine output of the diuretic group before time zero [i.e., on vasopressors, before receiving diuretics] and regardless of meeting AKI definition based on serum creatinine, those with oliguria would have had higher mortality rates [4, 5]. 

Indeed, based on the reviewer's suggestion, we have added the following notions to the results section in order to discuss competing risks and its effect on mortality.

Result section: “Prior to propensity matching, the median (IQR) of total urine output on the control group during the six hours after time zero (on vasopressors without diuretics) was 391 (186, 811) ml, while the median (IQR) urine output of the diuretic group during the 6 hours before time zero (on vasopressor before receiving diuretics) was 252 (133, 482) ml (P<0.001). In addition, using the competing risk model after accounting for the mortality, the sub-hazard risk ratio of AKI in the diuretic group in comparison with the control group was 1.65 (95% CI: 1.48-1.82, p<.001). After dividing the entire cohort based on the AKI status, defined as an increase in baseline serum creatinine by 1.5 times, as expected, the 90-day mortality of patients with AKI was significantly higher than those without AKI (OR 2.4, 95% CI 2.17, 2.68; P<0.001). Among those who did not receive diuretics, 415 (7%) and 504 (9%) received RRT in the first seven days and during the hospital admission, respectively. For the diuretic group, the number of patients who received RRT in the first seven days and during the hospital admission were 118 (13%), and 151 (16%) received, respectively, which was significantly higher than the control group (p<0.001).”

Discussion

• One of the limitations is the long study period. Clinical practice might have changed (e.g. the pattern of diuretics use as stated by the authors) during the study period and influence the results.

Authors’ Response: We agree this may represent a limitation. However, such changes in practice would equally affect both study groups. In addition, while practices have changed, the effect of diuretic on patients who are on vasopressors should not be different. In addition, when we added the year of the study as a variable in our models, it was not found to impact the outcomes significantly.

 

References:

[1] St Sauver JL, Grossardt BR, Leibson CL, Yawn BP, Melton LJ, 3rd, Rocca WA. Generalizability of epidemiological findings and public health decisions: an illustration from the Rochester Epidemiology Project. Mayo Clin Proc. 2012;87:151-60.

[2] Nelson JE, Cox CE, Hope AA, Carson SS. Chronic critical illness. American journal of respiratory and critical care medicine. 2010;182:446-54.

[3] Iwashyna TJ, Hodgson CL, Pilcher D, Bailey M, van Lint A, Chavan S, et al. Timing of onset and burden of persistent critical illness in Australia and New Zealand: a retrospective, population-based, observational study. Lancet Respir Med. 2016;4:566-73.

[4] Macedo E, Malhotra R, Bouchard J, Wynn SK, Mehta RL. Oliguria is an early predictor of higher mortality in critically ill patients. Kidney international. 2011;80:760-7.

[5] Prowle JR, Liu YL, Licari E, Bagshaw SM, Egi M, Haase M, et al. Oliguria as predictive biomarker of acute kidney injury in critically ill patients. Crit Care. 2011;15:R172.

---

## [Editor Report · Decision Letter 1]

13 Jan 2020

Use of Diuretics in Shock: Temporal Trends and Clinical Impacts in a Propensity-Matched Cohort Study

PONE-D-19-22578R1

Dear Dr. Kashani,

We are pleased to inform you that your manuscript has been judged scientifically suitable for publication and will be formally accepted for publication once it complies with all outstanding technical requirements.

With kind regards,

Chiara Lazzeri

Academic Editor

PLOS ONE
---

## [Editor Report · Acceptance letter]

7 Feb 2020

PONE-D-19-22578R1 

Use of Diuretics in Shock:  Temporal Trends and Clinical Impacts in a Propensity-Matched Cohort Study 

Dear Dr. Kashani:

I am pleased to inform you that your manuscript has been deemed suitable for publication in PLOS ONE. Congratulations! Your manuscript is now with our production department. 

With kind regards,

on behalf of

Dr. Chiara Lazzeri 

Academic Editor

PLOS ONE